# The Bone Regeneration Capacity of BMP-2 + MMP-10 Loaded Scaffolds Depends on the Tissue Status

**DOI:** 10.3390/pharmaceutics13070979

**Published:** 2021-06-29

**Authors:** Patricia Garcia-Garcia, Ricardo Reyes, José Antonio Rodriguez, Tomas Martín, Carmen Evora, Patricia Díaz-Rodríguez, Araceli Delgado

**Affiliations:** 1Department of Chemical Engineering and Pharmaceutical Technology, Universidad de La Laguna, 38206 La Laguna, Spain; pgarciag@ull.edu.es (P.G.-G.); cevora@ull.edu.es (C.E.); 2Institute of Biomedical Technologies (ITB), Universidad de La Laguna, 38320 La Laguna, Spain; rreyesro@ull.edu.es; 3Department of Biochemistry, Microbiology, Cell Biology and Genetics, Universidad de La Laguna, 38200 La Laguna, Spain; 4Laboratory of Atherothrombosis, CIMA, Universidad de Navarra, Instituto de Investigación Sanitaria de Navarra (IdisNA), 31009 Pamplona, Spain; josean@unav.es; 5Centro de Investigación Biomédica en Red en Enfermedades Cardiovasculares (CIBERCV), Instituto de Salud Carlos III, 28029 Madrid, Spain; 6Instituto de Productos Naturales y Agrobiología, CSIC, Francisco Sánchez 3, 38206 La Laguna, Spain; tmartin@ipna.csic.es; 7Instituto Universitario de Bio-Orgánica Antonio González, Universidad de La Laguna, Francisco Sánchez 2, 38206 La Laguna, Spain

**Keywords:** BMP-2, MMP-10, osteoporotic bone, microspheres, mesenchymal stem cells

## Abstract

Biomaterials-mediated bone formation in osteoporosis (OP) is challenging as it requires tissue growth promotion and adequate mineralization. Based on our previous findings, the development of scaffolds combining bone morphogenetic protein 2 (BMP-2) and matrix metalloproteinase 10 (MMP-10) shows promise for OP management. To test our hypothesis, scaffolds containing BMP-2 + MMP-10 at variable ratios or BMP-2 + Alendronate (ALD) were prepared. Systems were characterized and tested in vitro on healthy and OP mesenchymal stem cells and in vivo bone formation was studied on healthy and OP animals. Therapeutic molecules were efficiently encapsulated into PLGA microspheres and embedded into chitosan foams. The use of PLGA (poly(lactic-co-glycolic acid)) microspheres as therapeutic molecule reservoirs allowed them to achieve an in vitro and in vivo controlled release. A beneficial effect on the alkaline phosphatase activity of non-OP cells was observed for both combinations when compared with BMP-2 alone. This effect was not detected on OP cells where all treatments promoted a similar increase in ALP activity compared with control. The in vivo results indicated a positive effect of the BMP-2 + MMP-10 combination at both of the doses tested on tissue repair for OP mice while it had the opposite effect on non-OP animals. This fact can be explained by the scaffold’s slow-release rate and degradation that could be beneficial for delayed bone regeneration conditions but had the reverse effect on healthy animals. Therefore, the development of adequate scaffolds for bone regeneration requires consideration of the tissue catabolic/anabolic balance to obtain biomaterials with degradation/release behaviors suited for the existing tissue status.

## 1. Introduction

Bone tissue is under continuous remodeling throughout our lifetime. However, its regeneration capacity is limited by defects size, blood perfusion, age, and metabolic disorders [1]. Craniofacial bone defects or craniofacial congenital malformations generally require surgery and defects to be refilled with natural or synthetic biomaterials to promote bone healing. Often scaffolds are not enough to induce osteogenesis, and the contribution of other key elements such as progenitor cells or signaling molecules are required to ensure clinical success. In addition, the defects size and the comorbidity with other diseases altering bone tissue metabolism, compromise complete bone healing. Specifically, osteoporosis (OP) is characterized by an imbalance between the processes of bone formation and resorption and, consequently, by a delay in bone defects regeneration [2].

Bone morphogenetic proteins (BMPs) are a well-known growth factors family with osteo- and chondroinductive properties. Particularly, recombinant human BMP-2 (rhBMP-2) has shown a strong osteoinductive effect when incorporated in local delivery carriers placed in the injury site [3]. Postmenopausal osteoporosis is related to multiple factors, emphasizing the decreased levels of estrogens. These molecules play a key role in bone cell metabolism through estrogen receptor α (ERα). Estrogens inhibit osteoclastic bone resorption via increasing osteoclast apoptosis, reducing osteoblastic production of the receptor activator of nuclear factor ĸB ligand (RANKL) and increasing osteoprotegerin production [4]. Hence, in our previous works, scaffolds loaded with combinations of BMP-2 and 17β-estradiol formulated in microspheres were tested in an osteoporotic rat calvarial critical defect. Experimental results showed the new bone formed in OP rats was less mineralized than in non-OP rats [5,6,7]. On the other hand, bisphosphonates are routinely used drugs for OP treatment. These antiosteoclastic agents systemically administrated or integrated into scaffolds of different nature have been reported to support bone defects regeneration in both OP and non-OP animals [8,9,10,11,12]. Yet some authors stated no bisphosphonates effect in the regeneration of calvarial bone defects [13].

Moreover, BMPs have been combined with different bisphosphonates to reduce the excessive osteoclastogenesis that they induce as a consequence of their strong osteogenesis [14]. This combination formulated in local release systems generally causes a remarkable improvement in the bone defect regeneration induced by BMP-2 [15,16]. However, very few publications are devoted to studying the effect of this combination in osteoporotic animals. In fact, only two reports were found describing the effect of an IV or subcutaneous zoledronate single dose on the regeneration of a calvarial defect or a mid-diaphyseal open femoral fracture treated with a local release of BMP-2 or BMP-7, respectively [17,18].

The outcomes of both studies were different, the combination of a BMP-2 embedded collagen carrier with IV zoledronate initially had a negative effect in bone quantity and quality induced by rhBMP-2. Although this undesirable effect disappeared at long term assessments, the combination did not exceed the regeneration capacity of BMP-2 alone [17]. Conversely, the local implantation of BMP-7 incorporated in a collagen putty combined with zolendronate injected subcutaneously suggested that the hypothesis that osteoporosis is a disease that delays or hinders bone regeneration is incorrect [18]. However, the aforementioned studies are complicated to compare as defect models because the experimental designs and the BMPs used were different.

Lastly, matrix metalloproteinases (MMPs) are enzymes that play pivotal roles in tissue remodeling, by degrading a wide variety of matrix or nonmatrix substrates. MMPs expression, detected in osteoblasts and osteoclasts, has been shown to be required for bone homeostasis and proper fracture healing. In fact, altered cartilage remodeling and vascularization leading to impaired fracture repair has been reported in murine models with genetic deficiency in MMP9 [19] or MMP13 [20,21], while delayed bone remodeling has been observed in MMP2-KO mice [22]. Among them, MMP-10 has been shown to be expressed in osteoblasts and chondrocytes during bone formation in humans [23]. Interestingly, this MMP can enhance the in vitro osteoblastic differentiation of myoblastic cells induced by BMP-2 [24]. Moreover, we have recently demonstrated that MMP-10 is overexpressed in calcified human aortic valves (AVs) and active MMP-10 promotes calcification of valvular interstitial cells isolated from human AVs through Akt phosphorylation [25]. Furthermore, we have also shown that MMP-10 accelerates bone repair by enhancing BMP-2-promoted bone healing and improving the mineralization rate in a murine model of a calvaria critical size defect [26].

Therefore, the aim of this study is to further analyze the in vitro and in vivo MMP-10 pro-mineralization effect when combined with BMP-2 on a 3D scaffold. To accomplish this, different MMP-10 doses were tested. Moreover, the MMP-10 + BMP-2 activity was compared with scaffolds incorporating alendronate, a bisphosphonate, and BMP-2. Both proteins and bisphosphonate were formulated on microspheres and embedded on a crosslinked chitosan foam. The obtained foams were then surrounded by electrospun meshes loaded or not with alendronate to form the final scaffold with a sandwich-like structure. The in vitro promotion of ALP activity and in vivo mineralization using a critical size calvarial defect was assessed in both healthy and osteoporotic mice.

## 2. Materials and Methods

### 2.1. Microspheres Preparation and Characterization

#### 2.1.1. Microsphere Preparation

Different drug-loaded PLGA (poly(lactic-co-glycolic acid)) microspheres were designed for the experiment and prepared following different protocols (Table 1). PLGA microspheres loaded with BMP-2 and MMP-10 were prepared by a double emulsion method (*w*/*o*/*w*). For this purpose, a combination of two ester terminated PLGAs with variable molecular weight and lactide:glycolide ratio were selected. To obtain loaded microspheres 35 µg of BMP-2 (ED50: 0.74 µg/µL; GenScript, Piscataway, NJ, USA) and 2.1 µg or 8.5 µg rhMMP-10 (obtained equally to [27]) as described in Table 1 were dissolved in 200 µL of 0.8% PVA (MW: 30,000–70,000; Sigma-Aldrich, St. Louis, MO, USA). This aqueous phase was added to 1 mL of oil phase formed by the PLGA 75:25 (Resomer^®^ RG755, 0.54 dL/g, Evonik, Darmstadt, Germany) and 85:15 (Resomer^®^ RG858, 1.5 dL/g, Evonik) mixture used at 9 to 1 ratio at 150 mg/mL in DCM and homogenized for 1 min by vortex. Then, 3 mL of a second aqueous solution prepared with 2.5% PVA was added to the first emulsion and homogenized for one minute. Finally, the solvent was evaporated by pouring the obtained microspheres in 100 mL of PVA at 0.1% for one hour under continuous stirring.

Alendronate (ALD; Sigma-Aldrich) microspheres were prepared by a double emulsion method as described above with slight modifications. An internal aqueous phase (450 µL) was prepared containing PVA at 0.8%, chitosan (CHT, 150mPa.s, Protasan^®^ UP-CL-213, Sandvika, Norway) at 1%, and 75 µg ALD. The oil phase prepared as described above was added to this aqueous phase and homogenized for one minute. The second aqueous phase was added as described and the solvent was evaporated by pouring the microspheres in 100 mL of 0.1% PVA and 2.5% of Pentabasic sodium tripolyphosphate (STPP; Sigma-Aldrich) for one hour under continuous stirring.

The obtained microspheres were washed with double-distilled water, filtered through a 0.45 μm pore size filter (Pall Corporation, Sigma-Aldrich), freeze-dried, and stored at 4 °C until use. Microspheres were characterized in terms of size and size distribution by light diffraction using a Mastersizer (Mastersized 2000, Malvern Instruments, Malvern, UK) and the morphology was evaluated by scanning electron microscopy (SEM; JSM-6300, Jeol, Tokio, Japan) after silver coating.

#### 2.1.2. Microspheres Loading Efficiency

Protein loading efficiencies were evaluated using radiolabeled BMP-2 (125I-BMP-2) or MMP-10 (125I-MMP-10). Proteins were labeled following the Iodogen method [28] as previously described [29]. Briefly, in 50 µg Iodogen coated tubes (Pierce^®^ Pre-coated Iodination Tube, Thermo Scientific, Rockford, IL, USA) 50 µL of the protein solution (1 mg/mL) was mixed with 10 µL of 125INa (≈1 mCi) (Perkin-Elmer, Waltham, MA, USA) and taken to a final volume of 150 µL with Phosphate Buffer Saline (PBS, 0.5M pH 7). This mixture was kept at room temperature for 15 min under continuous stirring (120 rpm), then, 50 µL of a saturated tyrosine solution in PBS was added to remove any unreacted 125I. Labelled proteins were purified using ZebaTM Spin Desalting Column (Thermo Scientific).

ALD loading efficiency was evaluated on microspheres loaded with 99 mTechnetium radiolabeled ALD. Drug labelling was performed as described by Gundogdu and coworkers [30]. In Brief, 5 mg of ALD was mixed with 1 mg of ascorbic acid and 0.5 mg of tin chloride dihydrate (SnCl_2_ 2H_2_O) and kept under inert atmosphere. Afterwards, 0.5 mL of ultrapure water was added (MilliQ, Darmstadt, Germany). Once dissolved, 100 µL of generator eluate (0.5–1 mCi) was added and the solution was stirred (120 rpm) at room temperature for 15 min.

Labelling yields and stability were evaluated by instant thin layer chromatography (iTLC) using 11.5 × 0.8 cm silica gel coated strips (Varian Iberica SL) to which 5 µL of the labelled protein or ALD were added (30,000–40,000 cpm). For proteins, 85% methanol in water was used as mobile phase. A gamma counter (Cobra II, Packard^®^, Downers Grove, IL, USA) was used to measure free 125I (Rf = 1) and labelled proteins (Rf = 0) as previously published [31]. For ALD chromatography, acetone or 0.9% NaCl were used as mobile phase. The chromatography performed in acetone separated free 99mTc (99mTcO4-) (Rf = 1) from ALD-99mTc and hydrolyzed 99mTc (TcO_2_) (Rf = 0). On the other hand, when 0.9% NaCl is used as mobile phase both free 99mTc and ALD-99mTc migrate to the front (Rf = 1) while hydrolyzed 99mTc does not migrate (Rf = 0). These values can be used to calculate the percentage of labelled ALD as follows:%(^99m^Tc-ALD) = 100 − [% ^99m^TcO_4_^−^ + % ^99m^TcO_2_](1)

To evaluate the encapsulation efficiency of the microspheres, radioactivity was measured on the microspheres aliquots using a gamma counter (Cobra II, Packard^®^) and compared with total protein reactivity.

### 2.2. Preparation and Characterization of Electrospun Meshes

#### 2.2.1. PLGA Functionalization with ALENDRONATE

To obtain ALD-loaded electrospun fibers, two acid-terminated PLGA polymers with 50:50 lactide:glycolide ratio, Resomer^®^ RG502 (0.19 dL/g, Evonik), and Resomer^®^ RG504 (0.4 dL/g Evonik) were chemically conjugated with ALD following the protocol described by Choi and coworkers (Choi et al., 2007). A 10% PLGA solution was prepared in 20 mL acetone to which 60 mg of N-hydroxysuccinamide (NHS, Sigma-Aldrich) and 100 mg of N, N′-dicyclohexylcarbodiimide (DCC, Sigma-Aldrich) were added and allowed to react in inert atmosphere under stirring for 24 h at room temperature. Afterwards, the reaction mixture was filtrated through 0.45 µm (Chromafil^®^ Xtra PET-45/25, Macherey-Nagel, Lake Forest, CA, USA) to remove dicyclohexylurea and the obtained PLGA-NHS was precipitated in cold ethyl ether (Scharlau^®^, Barcelona, Spain) and dissolved in DCM. This solution was concentrated at room temperature until drying using a rotavapor.

ALD conjugation was performed by dissolving 1.9 g of the obtained PLGA-NHS in 19 mL of Acetone:DMSO (50:50) and adding an adequate amount of ALD previously dissolved in 1 mL of water. This mixture was allowed to react at room temperature for another 24 h. Then, the solution was concentrated by rotavapor at high vacuum and 37 °C. The obtained ALD conjugated polymer (PLGA-ALD) was dissolved in the minimum amount of acetone, precipitated in ethyl ether and dissolved again in DCM to avoid free ALD. Finally, this solution was dried for 24 h under high vacuum and the PLGA-ALD was precipitated with distilled water, filtrated through 0.45 μm, and freeze-dried. To ensure adequate functionalization, the obtained polymer was characterized by proton nuclear magnetic resonance. ^1^H-NMR spectra were recorded at 500 MHz (Bruker Avance 500, Billerica, MA, USA), and chemical shifts were reported in ppm and calibrated on non-deuterated solvent residual peak. Elemental analysis was performed using a CHNS TruSpec Micro analyzer (LECO, St. Joseph, MI, USA).

#### 2.2.2. Electrospun Meshes Preparation

Electrospun meshes composed by RG504-ALD: RG502-ALD: RG 855 at 1:1:1 ratio were prepared. The polymers mixture (375 mg) was dissolved in 2.5 mL of Hexafluoroisopropanol (HFIP, Sigma-Aldrich) and loaded into a syringe equipped with a 18G needle. This solution was ejected at a continuous flow of 2.75 mL/h using a syringe pump (Harvard Apparatus ^®^, Holliston, MA, USA) under an electric field of 7 KV. Fibers were collected on a cylindrical metal collector that rotates at 200 rpm located at a 10 cm distance from the syringe. The process took place at room temperature and 65% of relative humidity. Blank meshes with the same polymers proportions and conditions were also obtained using unfunctionalized PLGAs. Blank and ALD loaded electrospun meshes were surface treated with plasma (O_2_). The plasma oxygen treatment was carried out under vacuum during 4 min for each side of the meshes. The generator power was at 75% of the capacity (Diener electronic, Plasma-Surface-Technology, Ebhausen, Germany).

#### 2.2.3. ALD-Loaded and Blank Meshes Characterization

Meshes were characterized in terms of thickness, porosity, contact angle, and fiber diameter. Mean fiber diameter was obtained measuring 50 fibers for each sample after silver coating using scanning electron microscopy images (SEM, JSM 6300, JEOL) at a 1500 magnification by an image analysis software (ImageJ v1.52, National Institute of Health, Madison, WI, USA). Mesh thickness was obtained using stereomicroscope images (Leica M205 C, LAS, v3 software, Leica, Wetzlar, Germany).

Real mesh density was obtained by helium pycnometry (Micromeritics, AccuPyc 1330, Norcross, GA, USA) while apparent density was calculated from the weight and volume values. Volume was obtained from the length, width, and thickness as described in Equation (2). Mesh total porosity was calculated using real density and apparent density values as shown in Equation (3).
(2)$$Apparent\;density=\frac{weight}{\left(lenght\cdot width\cdot thickness\right)}$$
(3)$$Porosity\;\left(\%\right)=\frac{real\;density-apparent\;density}{real\;density}\cdot 100$$

Mesh surface wettability was measured by contact angle assessments using a drop shape analyzer (DSA100; Krüss GmbH, Hamburg, Germany) and distilled water as the liquid media. The obtained images were analyzed with an image analyze software (ImageJ v1.52, National Institute of Health).

A proton NMR spectrum was performed on a 500 MHz equipment to determine the amount of ALD present on ALD-loaded electrospun meshes. To do so, 4.41 mg of the sheet were dissolved in 0.6 mL of CDCl_3_, and 18.4 μL of a standard solution of 1,1,2,2-tetrachloroethane (Cl_2_CHCHCl_2_) in CDCl_3_ (33.07 mM) was added. In this way, the amount of Cl_2_CHCHCl_2_ was four times the theoretical amount of alendronate presented in the polymer sheet. The quantity of ALD was obtained by integrating the signals corresponding to 1,1,2,2-tetrachloroethane (5.96 ppm) and the alendronate signal of the protons (2.98 ppm). These measurements were performed also at different time points on ALD-loaded meshes incubated in water at 37 °C to evaluate ALD release. To this end, samples were washed twice with water and freeze-dried previous to NMR characterization.

### 2.3. Foams Preparation and Characterization

#### 2.3.1. Foam Preparation

Chitosan foams including adequate drug-loaded microspheres as shown in Table 2 were obtained. 4 mg of microspheres prepared under aseptic conditions as described in Section 2.1.1 were homogenously dispersed in 30 µL of a 3% chitosan solution, frozen at −20 °C and freeze-dried. Lyophilized systems were then crosslinked with 15 µL of 5% STPP, washed three times with 100 µL MilliQ water, frozen, and again freeze-dried. Afterwards, foams were stored at 4 °C until use. The porosity of the foam was calculated as described for electrospun meshes.

#### 2.3.2. Foams Wettability and Degradation

Foam water uptake and weight loss was analyzed by incubation in ultrapure water at 37 °C under continuous stirring (25 rpm). At different time points samples were removed, weighted after discarding the excess water and then freeze-dried to determine their mass loss.

#### 2.3.3. In Vitro Drug Release

Foams containing microspheres loaded with ^125^I radiolabeled proteins or ALD were incubated in sterile MilliQ water at 37 °C and 25 rpm. The amount of the released proteins was quantified by measuring the radioactivity of the supernatant with a gamma counter (Cobra^®^ II, Packard, Downers Gove, IL, USA) every other day. The released ALD, at the same time points, was measured using a derivatization method as previously reported [32]. Briefly, supernatants were reacted with o-pthaldialdehyde (OPA, Sigma-Aldrich) and 2-mercaptoethanol (Sigma-Aldrich). Then, the derivatization subproduct was measurement immediately using a spectrophotometer (Ultrospect 3300pro, Biochrom, Cambridge, England) (λ = 334 nm). In both cases, supernatants were removed at each time point and replaced by fresh media.

### 2.4. Osteoporosis Animal Model

Animal experiments were performed according to the European Union legislation on Care and Use of Animals in Experimental Procedures (2010/63/UE) and after approval by the Ethic Committee for animal care of the University of La Laguna (CEIBA 2014-0128, 5 November 2014).

The osteoporosis mice model was obtained by a surgical procedure consisting of a dorsal incision ovariectomy (OVX) under inhaled anesthesia (isofluorane, ISOFLO^®^, Abbott Laboratories, Valencia, Spain) on 16-week-old FVB mice. FVB mice were selected because other mice strains did not show signs of OP by densitometry. In addition, three weeks after ovariectomy, a 3 mg/kg dexamethasone 21-isonicotinate ((DEX) Deyanil Retard, Fatro Ibérica, Barcelona, Spain) was administered subcutaneously every week for three months. To confirm the development of the OP condition, bone mineral density (BMD) of the whole animal at different time points (0, 15, 30, 60 and 120 days post-OVX) was analyzed using a densitometer (PIXImus, GE Lunar, Madison, WI, USA). To perform the test, animals were anesthetized with medetomidine (1 mg/kg/IP) and ketamine (75 mg/kg/IP). To revert the anesthesia, atipamezole (0.1 mg/kg/IP) was administrated immediately after finishing the densitometry. Animal weight and height was monitored throughout the experiment.

Additionally, histological assessments of healthy and osteoporotic mice bone tissue four months after ovariectomy were performed. Mice were euthanized and femurs were extracted, fixed in 3.7–4% p-formaldehyde (PFA, pH 7) and decalcified in Histofix^®^ (Panreac, Barcelona, Spain). Samples were prepared for histological analysis as previously described [33].

### 2.5. Cell Isolation and Characterization

#### 2.5.1. Osteoporotic and Normal mMSCs Isolation

Normal and OP murine Mesenchymal Stem Cells (mMSC or OP mMSC) were isolated from tibia and femur of 8-week-old FVB healthy and osteoporotic mice, respectively, as previously described by Soleimani and Nadri with slight modifications [34]. Mice were sacrificed by cervical dislocation and both tibias and femurs were extracted. Muscle and periosteum were dissected and isolated bones were stored in Dulbecco’s phosphate-buffered saline (DPBS, Lonza, Merelbeke, Belgium) and processed for mMSC extraction. Both epiphyses were ruptured and a syringe equipped with a 20 G needle loaded with 10 mL of Dulbecco’s minimal essential medium (DMEM, Lonza) supplemented with 10% fetal bovine serum (FBS, Biowest, Riverside, CA, USA), 2 mM L-glutamime (Sigma-Aldrich), and 1% penicillin-streptomycin (Sigma-Aldrich) was used to flush the bone marrow to elute mMSC. This process was repeated twice for each bone. Eluates were then collected and centrifuged at 2000 rpm for 5 min, and the resulting pellets were combined, resuspended in 15 mL of DPBS, and centrifuged again under the same conditions. The resulting pellet was resuspended in 1.5 mL of complete DMEM and cells were seeded on a petri dish and incubated at 37 °C and 5% CO_2_. Cell culture media was changed every day for 72 h, then trypsinized (Trypsin-EDTA 0.25% in HBSS free of calcium, magnesium, and phenol red, Biowest) and seeded in cell culture flasks where they were allowed to grow until confluency.

#### 2.5.2. Characterization of Isolated Cells

The obtained normal and OP mMSC were characterized in terms of surface markers expression by flow cytometry analysis. Cells were cultured until passage 3, then trypsinized and stained with calcein AM (1 µM; Sigma-Aldrich) for 30 min at room temperature. Afterwards, cells were washed three times with DPBS divided into 4 tubes and incubated with the correspondent APC-labelled primary antibody for CD45, CD44, and Sca-1 or APC-labelled isotype control (ThermoFisher Scientific) for 30 min at 4 °C. The concentration used for both control and specific antibodies was the same 10 µg/mL. Then, cells were washed three times with DPBS and stored at 4 °C until characterization. Flow cytometry was performed using the Macsquant Analyzer 10. Untreated cells were used as control.

### 2.6. In Vitro Biological Performance of Chitosan Foams and Electrospun Meshes

#### 2.6.1. Evaluation of Developed Foams Osteogenic Capacity

Alkaline phosphatase activity was used to evaluate the osteogenic and mineralization effect of the drug-loaded chitosan foams described in Table 2 on normal and OP mMSC. Experiments were performed using isolated murine MSCs. Cells were trypsinized and resuspended at high density 1 × 10^6^ cells/mL in complete media. Then, 20 µL of the cell suspension was added to the foams and incubated for 1.5 h at 37 °C and 5% CO_2_ to increase cell attachment. Afterwards, 500 µL of complete medium was added to each well and media was changed every other day. After 7 and 21 days of culture, samples were collected in triplicate to evaluate osteogenic differentiation. Foams were washed twice with DPBS at 4 °C and 500 µL of the developer solution was added and incubated at 37 °C and 5% CO_2_ under gentle agitation for 1.5 h. The developed solution was a solution of tetrazolium nitro blue chloride (NBT, Roche Diagnostics, Mannheim, Germany) and 5-bromo-4-chloro-3-indole phosphate (BCIP, Roche Diagnostics) prepared in 0.1 M Tris-HCl, 0.1 M NaCl, and 0.05 M MgCl2 (pH = 9.2–9.5). After incubation, samples were washed twice with DPBS and fixed with buffered PFA. After 30 min fixation at room temperature, foams were washed twice, dehydrated, included in paraffin (Paraplast^®^, Barcelona, Spain) and cut with the microtome (5 µm thickness; Shandon Finesse 325, Thermo Fisher Scientific). Images of the obtained sections were taken in a microscope (Leica DM4000B, Barcelona, Spain). The percentage of cells stained positive for alkaline phosphatase activity was obtained comparing the number of positive cells against total cell numbers using an image software (ImageJ v1.52, National Institute of Health).

#### 2.6.2. Evaluation of Cell Viability and Adhesion to Electrospun Meshes

OP mMSC viability and adhesion to the plasma-treated electrospun meshes was evaluated. Electrospun meshes were cut in disks of 8 mm diameter and placed on 48-well plates. Then, 100 µL of the cell suspension (2.5 × 105 cells/mL) was added to the meshes and incubated for 1.5 h at 37 °C and 5% CO_2_ to promote cell adhesion followed by the addition of 400 µL of complete media to each well. After 24 h of seeding, cell viability was evaluated by the XTT assay (XTT Cell Viability Kit, Cell Signaling Technology, Indianapolis, IN, USA) following the manufacturer’s instructions. Absorbance was measured at 445 nm using a plate reader (Biotek, Winooski, VT, USA). Cells seeded on tissue culture treated polystyrene were used as control (100% viability).

To study cell adhesion, cells were seeded and incubated for 24 h with the meshes as described above and then fixed with PFA. To evaluate cell morphology, cells were stained with rhodamine phalloidin (Invitrogen, Eugene, OR, USA) and DAPI. Samples were washed twice with DPBS and permeabilizated with 0.1% Triton X100 in DPBS for 15 min. Then, samples were washed twice again with DPBS and phalloidin solution; 1% BSA was added and then incubated at room temperature for 40 min. Afterwards, samples were washed twice with DPBS and 1 µg/mL DAPI was added and incubated for 5 min at room temperature. Finally, samples were washed twice with DPBS, mounted with Fluromount (Invitrogen) and observed under a fluorescence microscope (Leica DM4000B, Leica Microsystems, Barcelona, Spain). Images were acquired using a digital camera (Leica DFC300FX). Cells per field were counted using an image analysis software (ImageJ v1.52, National Institute of Health).

### 2.7. In Vivo Experiments

#### 2.7.1. Surgical Procedure

For this study, 16-week-old FVB female mice weighing between 25–35 g were used. Mice were divided into two groups: healthy mice (non-OP group) and osteoporotic mice (OP group) as described in Appendix A. After 4 months, both mice groups were subjected to a surgical procedure under inhalation anesthesia in which a 4 mm diameter critical bone defect was performed in the calvaria. Prior to the surgical intervention, the animal’s head hair was shaved and povidone iodine was applied, a sagittal incision was made along the skull, and the skin was displaced to leave the skull exposed. Critical bone defects were made following a previously published protocol [35]. Electrospun meshes containing ALD for the BMP-2+ALD foam group (Table 3) or blank electrospun meshes for the remaining groups (Control, BMP-2 foam, BMP-2 + Low MMP-10 foam and BMP-2 + High MMP-10 foam; Table 3) were placed on the defect. Adequate foams were then placed above the meshes and the defect was closed placing another electrospun mesh with or without ALD, depending on the treatment group, on top of the foams. Finally, the wound was closed with surgical staples. Analgesia consisted of buprenorphine administered subcutaneously (0.01 mg/kg) before surgery and paracetamol (200 mg/kg) in water for 3 days after intervention. Both non-OP and OP groups were subjected to 4 different treatments (10 mice each) as shown in Table 3. An additional control group (10 mice) including control foam and blank electrospun mesh was added to the non-OP group as control of bone formation and mineralization without any drug treatment. Therefore, the experiment was performed using 90 mice: 50 non-OP and 40 OP. Two time points were tested, 6 and 12 weeks, and 5 mice were used for each time point. Animals were euthanized by CO_2_ inhalation. During all the experiments, both OP and non-OP mice were monitored for bone mass loss by densitometry (PIXImus, GE Lunar) as described in Section 2.4.

#### 2.7.2. In Vivo Drug Release

To evaluate the in vivo protein delivery, foams loaded with radiolabeled BMP-2 (125I-BMP-2) or MMP-10 (125I-MMP-10) were used. The release rate was evaluated by measuring the amount of radiolabeled proteins remaining in the scaffold at different time points using a non-invasive method. The signal was compared with the one obtained immediately after implantation as previously described [35,36]. The radioactivity was measured using a probe-type gamma counter (Captus^®^, Nuclear Iberica, Ramsey, NJ, USA) coupled to a 3.2 × 2 cm collimator. Mice were immobilized and 3 measurements (27KV, 1 min) were performed. The average of the measurements carried out just after implantation was considered as the administered dose and the release percentage was obtained as the difference between the administered radioactivity and the remaining radioactivity at each time point, all of which was corrected by the tracer disintegration factor, in this case 125I (t1/2 = 60 days).

Moreover, for MMP-10 release experiments, an extra group consisting of plain MMP-10 microspheres (no foam and no mesh) was added. In this group, 7 µL of 15% Pluronic PF127 was added to ensure microspheres stay at the defect site [26].

#### 2.7.3. Histological and Histomorphometric Evaluation

To label the mineralization front, animals were injected with oxytetracycline-HCl (40 mg/kg, IM) and calcein blue (15 mg/kg, SC) 12 and 4 days previous to euthanasia, respectively.

To assess the in vivo effects of the different treatments, histological and histomorphometric assessments of the samples were performed. After tissue fixation in 10% formalin solution (pH 7.4), undecalcified bone specimens were prepared for histological analysis as previously described [37]. The sections were stained with Goldner’s trichrome to identify new bone formation, or left unstained for detection of fluorochrome labels, and analyzed by light microscopy (LEICA DM 4000B, Barcelona, Spain).

For histomorphometrical analysis, all sections per specimen were evaluated using computer-based image analysis software (Leica Q-win V3 Pro-image analysis system, Barcelona, Spain). We defined a region of interest (ROI) consisting of a circular area of 12.5 mm^2^, the center of which coincided with that of the defect site. This region covered the entire defect surface and was limited by the host bone. Within this ROI, newly formed bone was distinguished from scaffold material through structure and color differences. New bone formation was expressed as a percentage of repair in relation to the total area of the defect. The distance between tetracycline and calcein blue labels was measured under ultraviolet light for the calculation of mineral appositional rate (MAR).

### 2.8. Statistical Analysis

Statistical analysis was performed with SPSS version 25 software. All experiments were run at least in triplicate. Differences between the treatment groups were analyzed by a one-way analysis of variance (ANOVA) with a Tukey multiple comparison post-test. For the in vivo studies, different treatments at each time point (6 weeks and 12 weeks) were compared by means of a two-way ANOVA with a Tukey multiple comparison post-test. Significance was set at *p* < 0.05. Results are expressed as mean ± SD.

## 3. Results

### 3.1. Physicochemical Performance of Developed Systems

#### 3.1.1. Polymeric Microspheres

Polymeric microspheres were obtained by a double emulsification method using an initial aqueous internal phase of either PVA+H2O (PLGA microspheres) or PVA+Chitosan+H2O (PLGA/Chitosan microspheres). Both types of microspheres showed similar mean diameter being 69.18 ± 1.6 µm for PLGA microspheres and 65.98 ± 0.04 µm for PLGA/Chitosan microspheres with a single distribution peak. Microspheres’ loading efficiencies were obtained using radiolabeled proteins or ALD. PLGA microspheres showed high encapsulation efficiencies for BMP-2 (70 ± 6.8%) and MMP-10 (74.05 ± 25.9%). On the other hand, the obtained alendronate encapsulation efficiency was slightly lower, 63.11 ± 11.02%.

#### 3.1.2. Electrospun Meshes

To obtain electrospun meshes incorporating ALD, the drug was covalently linked to two different PLGAs. 1H NMR study (Appendix A) was performed to confirm the conjugation. The chemical shift (δ), expressed in ppm, gives information on the type of hydrogen generating the signal. When the lower molecular weight PLGA was used to link ALD, Resomer^®^ RG 502, chemical shifts were observed at 1.57 (m, 321 H), 2.98 (s, 2 H), 4.54–4.98 (m, 183 H) and 5.07–5.38 (m, 91 H). On the other hand, when Resomer^®^ RG 504 H was used as the polymer backbone for drug incorporation signals at δ 1.57 (m, 352 H), 2.97 (s, 2 H), 4.54–4.98 (m, 246 H) and 5.07–5.38 (m, 114 H) were recorded. Integrating the signals obtained from PLGA (5.25–5.16 ppm) and the alendronate signal at approximately 3 ppm, the obtained percentage of polymer functionalization was 76.3% for Resomer^®^ RG 502 and 98.8% for Resomer^®^ RG 504. Moreover, the percentage of RG 502-ALD and RG 504-ALD polymers over the theoretical amount on the final mesh was determined in triplicate by comparing the integrals of Cl_2_CHCHCl_2_ (used as internal standard) and the alendronate signal at 3 ppm correspondent to “a”, a (R-CH2-X) proton X being the Nitrogen (N) of ALD (Figure 1). The obtained value was 89 ± 7% indicating no loss of the linked ALD during storage and electrospinning. Therefore, the ALD dose on the meshes was 7 ng per mg of electrospun mesh. On the other hand, samples soaked for 72 h in water did not show any ALD signal (Appendix A) indicating the release of the drug at this time point. These effects could be more related to polymer hydrolysis than to the hydrolysis of the amide group that is formed in the conjugation of ALD with the polymer.

The obtained sheets were afterwards treated with plasma to decrease the surface hydrophobicity conferred by PLGA. Indeed, blank-treated meshes showed contact angle values around 90°, the hydrophobic materials threshold, (88.2 ± 10.7) while ALD-meshes showed a slightly higher mean contact angle of 93.43 ± 8.17. However, both types of meshes presented lower contact angles than those previously reported for non-treated PLGA meshes 114.1 ± 8.3 [38].

Furthermore, the physical properties and morphology of the meshes were characterized in terms of thickness, fiber diameter, and total porosity (Figure 2). The obtained blank and ALD-electrospun fibers presented a smooth surface (Figure 2A,B) characterized by a similar microscale fiber diameter (≈1.3 µm). As expected, both systems are highly porous showing a total porosity above 80%. However, the thickness of ALD meshes was significantly higher than the one for blank meshes (Figure 2C).

#### 3.1.3. Microsphere-Loaded Chitosan Foams

Chitosan foams including adequate microspheres based on the desired scaffold composition were obtained by ionic crosslinking using a straightforward approach. The obtained foams were characterized by high porosity, 94.6 ± 1.2%, and water uptake with 366.5 ± 13.9% of swelling after 24 h (Figure 3A). Moreover, the obtained foams showed high stability with a mass loss of only 10.9 ± 5.13% after 28 days of study. Interestingly, the mass loss was stable throughout the experiment being 15.1 ± 5.7% just after 24 h.

On the other hand, the in vitro release studies showed a diffusion-controlled release profile for both of the incorporated proteins reaching around 70% of protein release, 69.98 ± 2.62% for BMP-2 and 83.28 ± 4.21% for MMP-10, after 18 days of study (Figure 3B). However, ALD showed a more pronounced burst release with 55.41 ± 3.21% of the drug released just after two days of study followed by a sustained release rate. Despite the high burst release effect observed, the incorporation of ALD microspheres into the chitosan matrix was able to decrease the burst effect showed by plain microspheres characterized by almost all the drug released just after 2 days (94.4 ± 5.6%) (Appendix A).

### 3.2. Osteoporosis Instauration

To obtain OP-like mMSC and to evaluate the in vivo scaffold performance in OP-like environments, an osteoporotic mice model was developed. Both healthy and OP mice showed normal growth. Densiometry results (Figure 4A) showed significantly lower bone mineral density values for OP-mice two weeks after ovariectomy when compared with control animals (*p*-value < 0.001). From then, this difference was maintained during the remaining time of the experiment. Moreover, the histomorphometric assay results confirm the OP condition (Figure 4B). Femurs from OP mice showed significantly lower cortical bone thickness, width and number of trabeculae in cancellous bone, and higher separation of trabeculae in cancellous bone.

### 3.3. Characterization of OP-Like mMSC and “Healthy” mMSC

Cell surface markers expression was analyzed for OP-like mMSCs and control/healthy mMSC to validate the isolation procedure selected. Flow cytometry results (Figure 5) showed the expected mMSC surface markers expression with positive expression of mMSC identifiers markers; stem cell antigen-1 (Sca-1) and CD44 and no expression (OP mMSC) or slight expression (non-OP mMSC) of hematopoietic markers (CD45).

### 3.4. In Vitro Performance of Chitosan Foams and Electrospun Meshes

Alkaline Phosphatase (ALP) is a crucial enzyme on the biomineralization process increasing the local concentration of inorganic phosphate and a good predictor of neotissue mineralization [39]. Both non-OP and OP mMSCs were cultured on microsphere-loaded chitosan foams to analyze whether the presence of BMP-2 alone, or combined with MMP-10 or ALD, successfully promoted MSC osteogenic differentiation and mineralization in both cell populations. After 7 and 21 days, ALP activity was evaluated and foams were fixed, cut, and mounted on slides. The microscopic analysis showed the presence of positively stained cells (ALP + cells) in all the experimental groups at the two time points were analyzed. The level of ALP activity in most cells is qualitatively high as shown by the intensity of the labelling (Figure 6A,B); however, slight qualitative differences were observed between mMSC from normal mice (non-OP mMSC) and mMSC from OP mice, observing a higher labelling intensity in the first ones (Figure 6A,B).

Quantification showed a significant increase in the number of non-OP mMSC ALP + cells at 7 days of culture in all the treated groups with respect to the control group, which was highest when cells were cultured on chitosan foams combining BMP-2 and ALD and BMP-2 and MMP-10 at low amounts (Figure 6C). In fact, cells grown on these foams showed significantly higher ALP activity than those cultured on systems including BMP-2 alone. The same trend was observed after 21 days of culture where drug-loaded foams showed significantly higher number of ALP + cells than blank foams. Moreover, the combination of BMP-2 with ALD or MMP-10 on the foams led to an increased ALP activity when compared with the addition of BMP-2 alone.

On the other hand, this beneficial effect of the mixture of BMP-2 and ALD or MMP-10 was not observed on OP mMSC ALP activity (Figure 6D). In these circumstances, the incorporation of BMP-2 to chitosan foams successfully increased the number of ALP + cells after 7 days of culture but no additional effect was observed by its combination with ALD or MMP-10. Moreover, after 21 days of culture, only the treatment combining BMP-2 and low dose of MMP-10 presented a significantly higher number of ALP+ cells compared with control foams. In general, foams seeded with OP mMSC, at 7 and 21 days post-seeding, showed a lower number of ALP+ cells per field than those seeded with non-OP mMSC. Therefore, a clear difference in mineralization potential can be observed between the control and OP mMSC.

Moreover, OP mMSC were cultured on electrospun PLGA meshes containing ALD or not. Cell viability and attachment were evaluated to analyze the effect of ALD presence on cell behavior. Interestingly, the presence of ALD on the meshes significantly improved cell viability (Figure 7A). Fluorescence images confirmed this trend (Figure 7B,C), showing a higher number of attached cells on ALD-linked electrospun meshes.

### 3.5. In Vivo Evaluation of Sandwich-Like Scaffolds

#### 3.5.1. In Vivo Protein Drug Release

The in vivo release of BMP-2 and MMP-10 incorporated in chitosan scaffolds showed similar profiles to those observed in vitro but with a slight increase on the burst effect (Figure 8). The amount of protein released after 24 h of implantation represented the 22.32 ± 2.68% of the dose for BMP-2 and 25.51 ± 2.22% of the dose for MMP-10. The release rate of both proteins fit the Higuchi model, characteristic of release processes controlled by diffusion also explained by the high chitosan foam stability previously observed. The Higuchi constant (KH) was 13.05 for BMP-2 release (R^2^ = 0.99) and 11.31 for MMP-10 (R^2^ = 0.97). Therefore, the complete dose of BMP-2 and MMP-10 would be released after 8 weeks and 11 weeks, respectively, indicating the achievement of a stable prolonged release.

#### 3.5.2. In Vivo Bone Formation Induced by Sandwich-Like Scaffold Containing BMP-2 Alone or Combined with ALD or MMP-10

The ability of the developed sandwich-like scaffolds to promote new bone formation and tissue mineralization was evaluated on a critical size calvarial bone defect on both normal (non-OP) and OP animals. When scaffolds were implanted on control “healthy animals” a high proportion of connective tissue surrounding the scaffold and low new bone formation restricted to the margins of the defect was observed for all experimental groups (Figure 9A). Twelve weeks post-implantation, a significant increase in the repair response was observed with newly formed bone at the margins of the defect and isolated ossification foci within it (Figure 9). The histomorphometric analysis showed significantly higher repair percentages for all the treatments compared with control scaffolds (data not shown) except for the combination of BMP-2 and MMP-10 with a high dose of the MMP. Both BMP and BMP + ALD treatments showed similar percentages of repair at both 6 weeks (≈30%) and 12 weeks (≈50%) post-implantation. Conversely, the combination of BMP+MMP at the two concentrations tested led to a significant decrease in the percentage of repair at both time points when compared with both BMP and BMP +ALD. The combination of 120 ng MMP-10 and BMP-2 was only able to achieve 27% of bone repair after 12 weeks of implantation (Figure 9C).

The mineralization rate, determined by the mineralization fronts marked with calcein blue and oxytetracycline, revealed significantly lower mineral apposition rates (MAR) at six weeks post-implantation than at twelve weeks for all the experimental groups (Figure 9D). Contrary to the results in bone repair, the combination of BMP-2 and MMP-10 led to a significant increase in mineral apposition rates compared with BMP-2 alone. In fact, when the higher dose of MMP-10 was used, 120 ng, the mineral apposition rate was significantly higher to that observed for both BMP and BMP + ALD treatment groups.

Scaffold’s performance was also evaluated on the developed OP mice model. In osteoporotic animals the relative proportion of connective tissue in the area of the defect was similar to that observed in normal animals (Figure 9B). Following the same trend as for control animals, the obtained percentage of repair and mineral apposition rates were significantly higher at the last time point studied (12 weeks) than at 6 weeks post-implantation (Figure 9C,D). At this point of analysis, new bone formation was observed not only in the margins but also in larger areas of the defect. However, a shift in the MMP-10 effect on tissue repair in OP animals was clearly observed when compared with control non-OP animals. In diseased animals, the combination of BMP-2 and MMP-10 at either of the doses selected promoted a significant increase in tissue repair compared with BMP-2 alone and BMP + ALD (Figure 9C). The highest percentages of repair were observed on mice implanted with BMP + MMP-L with a value of 44% after 6 weeks of implantation and 54% after 12 weeks of implantation. On the other hand, the lowest percentages of repair were obtained with scaffolds containing plain BMP showing values of 17% after six weeks of implantation and 35% after 12 weeks.

In this case, the results of mineral apposition rate match the results mentioned above for tissue repair (Figure 9D). The combination of BMP-2 and MMP-10 at either of the doses selected promoted a significant increase in the mineral apposition rate compared with the use of scaffolds including BMP-2 alone or combined with ALD.

## 4. Discussion

The development of biomaterials designed for bone regeneration has been one of the main objectives in the tissue engineering field for the last decades. The strategy of these scaffolds has been mostly centered on fulfilling requirements related to scaffold mechanical performance and stem cell osteogenic induction capacities recognized as crucial to ensure the recovery of the bone function. However, not only should the characteristics of the scaffold be considered but the quality of the regenerated bone should also be taken into account, especially when scaffolds are used in pathologies characterized by an impaired bone quality. Increased bone mineral content and improved microarchitecture is directly correlated with enhanced mechanical properties and is desirable for osteoporosis treatments [40]. In this study we propose to use already well-known biocompatible and biodegradable raw materials to develop sandwich-like scaffolds designed to obtain better bone quality, increasing the mechanical resistance of bone. To do so, scaffolds were doped with BMP-2 alone, with known osteogenic induction capacities, or with combinations of BMP-2 and an antiresorptive drug (ALD), or BMP-2 and metalloprotease 10 (MMP-10), at different doses.

To achieve a desirable growth factor and ALD or MMP-10 controlled release while avoiding the risk of undesirable side effects derived from high concentrations, therapeutic molecules were included on PLGA microspheres. The developed microspheres were characterized by high encapsulation efficiencies of over 60% in all cases, being even 10% higher for BMP-2 and MMP-10. The addition of chitosan in the internal phase during the synthesis of ALD-loaded PLGA microspheres aimed at increasing their drug encapsulation capacity which is a challenge due to the high aqueous solubility of the drug. Indeed, the developed microspheres presented a five-fold increase in ALD encapsulation efficiency compared with previously reported ALD-loaded PLGA microspheres prepared following a similar double emulsification method [41]. Moreover, other reports already reported lower ALD encapsulation efficiencies using a double emulsification method with levels as low as 0.2% [42].

Microspheres were then incorporated into chitosan networks serving as molds to obtain cross-linked foams with high water uptake capacity, porosity, and remarkable stability. Chitosan is a natural polymer widely used for bone tissue engineering applications due to its several advantages as easy chemical modification and high biocompatibility [43]. Plain chitosan foams have been previously tested as substrates for osteoblasts attachment indicating adequate performance [44,45]. However, chitosan is commonly combined with either ceramics and/or other polymers to improve mechanical performance and confer osteoinductive capabilities [46]. The strategy used on this study proposes to include microspheres loaded with BMP-2 and MMP-10 or ALD into chitosan foams serving as a therapeutic molecules’ reservoir and conferring osteoinductive capabilities to the developed systems. In agreement with our hypothesis, the incorporation of the microspheres to chitosan foams led to a controlled release of the therapeutic molecules both in vitro and in vivo characterized by a diffusion-controlled profile. Moreover, the incorporation of PLGA microspheres to chitosan foams could improve the mechanical performance of the systems.

The ability of the incorporated therapeutic molecules to confer osteoinductive capabilities to the developed foams was evaluated by their culture with MSC from “healthy” and OP animals and then the alkaline phosphatase activity was analyzed. The testing of scaffold performance using both cell populations is justified by the known alterations in OP MSC [47]. Cell populations were characterized by flow cytometry showing different profiles for Sca-1 and CD44 expression. Sca-1-deficient mice were previously reported to undergo osteoporosis and are characterized by a decreased self-renewal of osteogenic stem cells [48,49,50]. In fact, the administration of Sca-1 + sorted MSC to osteoporotic animals was able to improve bone mineral density [51]. On the other hand, similarly to Sca-1, CD44 seemed to be decreased in OP mMSC. The absence of this cell surface hyaluronan receptor has been correlated to increased osteoclastogenesis and bone loss under inflammatory environments in mice [52]. Moreover, CD44 has been described as crucial for MSC migration, a cell function reported to be decreased in OP mMSC [53,54]. Altogether, the modified positive expression of Sca-1 and CD44 observed for OP mMSC (Figure 5) could be associated to the disease instauration and the associated lower MSC migration, proliferation, and differentiation capacity [47,54]. Indeed, OP mMSC showed a lower number of ALP+ cells than non-OP mMSC for all the evaluated treatments at both of the time points selected. Analyzing the response of both cell populations to BMP-2 alone or combined with the other therapeutic molecules, clear differences were pointed out. The addition of BMP-2 significantly increased the number of ALP+ cells for both populations, but while the combination of BMP-2 + ALD and BMP-2 + MMP-10 at low dose produced an even higher number of ALP + cells for non-OP cells, no effect was observed for OP mMSC. The ability of MMP-10 to increase the number of ALP + cells under an osteogenic environment was previously described in vitro on aortic valve interstitial cells [25]. This effect has been hypothesized to be mediated mainly by the PI3K/AKT cell signaling pathway, a critical signaling pathway involved in OP [55]. This fact could explain the differences in cell response to MMP-10. On the other hand, in agreement with our findings for non-OP mMSC, ALD was previously reported to promote osteogenesis and an enhancement in ALP expression on human MSC [56].

A new strategy was used to load PLGA electrospun meshes with ALD. In this case, the drug was efficiently conjugated to different PLGA polymer chains. Thus, the obtained meshes presented a drug content directly controlled by the quantity of functionalized polymers used for electrospinning and not affected by storage. Plasma-treated electrospun meshes presented high porosity, microscale fiber diameter and, therefore, a high surface-area-to-volume ratio, which are adequate characteristics to resemble the natural extracellular matrix [57]. Indeed, osteoporotic-like cells seeded on the developed meshes showed good attachment improved by the presence of ALD on their surface. These differences could be attributed to the acid character of un-functionalized electrospun meshes due to the carboxylic acid terminal group of non-modified PLGA.

To obtain suitable scaffolds for bone regeneration, the developed chitosan foams were surrounded by two electrospun meshes generating a 3D sandwich-like scaffold. To the best of our knowledge, no previous work has been devoted to combine BMP-2 and MMP-10 on microspheres embedded in a scaffold for the regeneration of critical size bone defects in OP animals. Furthermore, the results of this combination were compared with scaffolds containing BMP-2 alone or BMP-2 + ALD. In our previous work, microspheres loaded with BMP-2 and MMP-10 were dispersed in a hydrogel and implanted on a critical size bone defect [26]. The regenerative effect of the hydrogels was evaluated after 4 and 8 weeks of implantation and the ratios tested for BMP-2: MMP-10 were 20:1 and 200:1. The obtained results suggested a higher percentage of bone repair and mineralization apposition rates for the higher MMP-10 dose (20:1; 30 ng MMP-10) after 4 weeks of implantation [26].

As described above, bone formation is impaired in OP. Moreover, our previous results revealed that a slow release rate of BMP-2 can be beneficial for bone regeneration promotion in OP animals. Furthermore, the development of solid scaffolds able to remain longer on the defect site leads to better bone repair [58]. Therefore, in the present study, scaffolds were designed to persist in the defect site for a long time and to slowly release the therapeutic molecules. In fact, the presence of the scaffolds on the defect area was evident at both six and twelve weeks of implantation. Likewise, a complete infiltration of the scaffold by connective tissue was observed, with the presence of cells both between the microspheres and inside them. However, this long-term stability of the scaffolds had a negative effect on non-OP animals, hindering the rapid regeneration observed on these animals and leading to lower regeneration percentages to those previously reported for BMP-2 + MMP-10 [26]. This fact could also explain the in vivo lack of positive response in percentage of repair for the BMP+MMP and BMP+ALD treatments compared with plain BMP-2, despite the beneficial effects in ALP activity observed for these treatments in vitro. Moreover, the dose-dependent negative effect of MMP-10 addition on the regenerative response could be attributed to a catabolic effect aimed to degrade the scaffold. Despite the decrease observed in tissue repair, the mineralization rate for BMP+MMP treatments was significantly higher than BMP-2, confirming the positive effect of MMP-10 addition on bone mineralization.

The results obtained suggested the designed sandwich-like scaffolds are a closer match for the bone tissue repair requirements in OP animals. In these animals, the combination of BMP-2 and MMP-10 induced a regenerative effect and an improved mineral apposition rate when compared with scaffolds containing BMP-2 alone or BMP-2 + ALD. These observations suggested the requirements to achieve a good bone regeneration in the osteoporotic population are different than those in healthy animals. Therefore, for bone regeneration it is clear that there are three factors that must be controlled as well as their interactions to achieve the desired effect: (1) the activity of the bone tissue to be repaired; (2) the time the scaffold stays in the defect or, in other words, the rate of scaffold degradation and; (3) the release rate and dose of the active substances as well as their combinations. Future work is necessary to improve the understanding of the underlying interactions between these three factors to exploit them and guide the development of novel therapeutic strategies for bone regeneration.

## 5. Conclusions

Sandwich-like scaffolds were obtained combining PLGA electrospun meshes and drug-loaded chitosan foams. Therapeutic molecules were efficiently loaded and released from microspheres embedded on the chitosan network acting in vivo and in vitro as therapeutic molecule reservoirs for the developed scaffolds. This versatile strategy for scaffold development allowed testing of the effect of BMP-2 and BMP-2 + MMP-10 or BMP-2 + ALD on osteogenic induction and bone regeneration. The obtained results point out the crucial role of the bone tissue repair activity on the treatment’s success. The designed BMP-2 + MMP-10 combinations showed the desired effect of tissue repair promotion on OP animals, but the opposite effect was observed on control animals. Therefore, the development of adequate scaffolds for bone regeneration require consideration of the tissue catabolic/anabolic balance to obtain biomaterials with a degradation/release behavior suited for the existing tissue status.

## Figures and Tables

**Figure 1 pharmaceutics-13-00979-f001:**
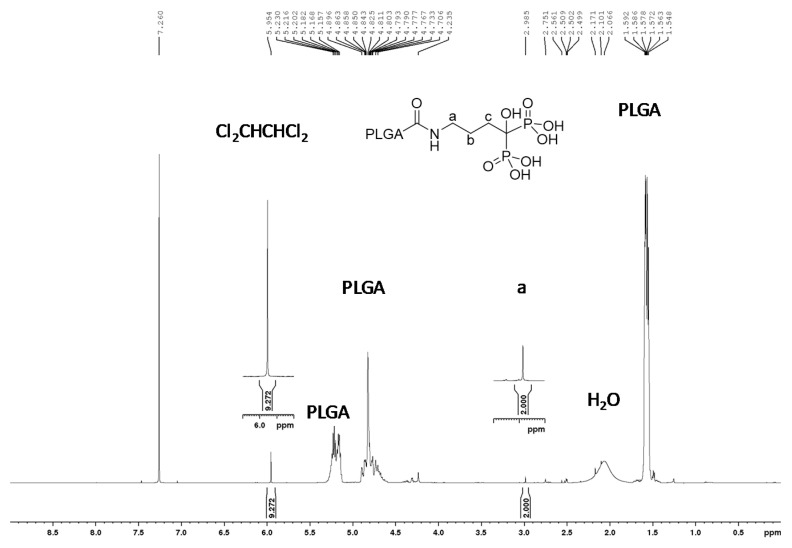
1H-NMR spectrum of electrospun meshes soaked using Cl_2_CHCHCl_2_ as internal standard to quantify the amount of PLGA-ALD. The signal at 7.26 ppm corresponds to the solvent signal CHCl_3_.

**Figure 2 pharmaceutics-13-00979-f002:**
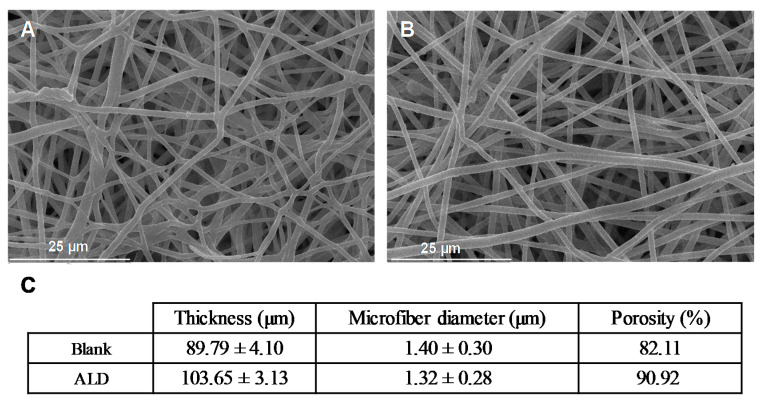
Scanning electron microscopy images at 1500× magnification of: (**A**) blank meshes (without ALD) and (**B**) alendronate meshes both treated with plasma oxygen. (**C**) Thickness, microfiber diameter, and porosity of the blank and ALD linked meshes.

**Figure 3 pharmaceutics-13-00979-f003:**
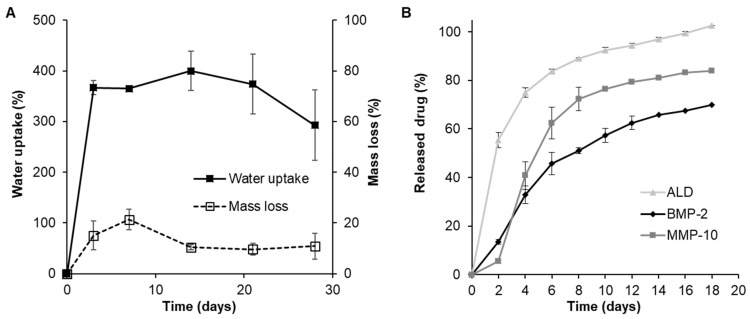
(**A**) Water uptake and mass loss and (**B**) drug release profiles of microsphere-loaded chitosan foams.

**Figure 4 pharmaceutics-13-00979-f004:**
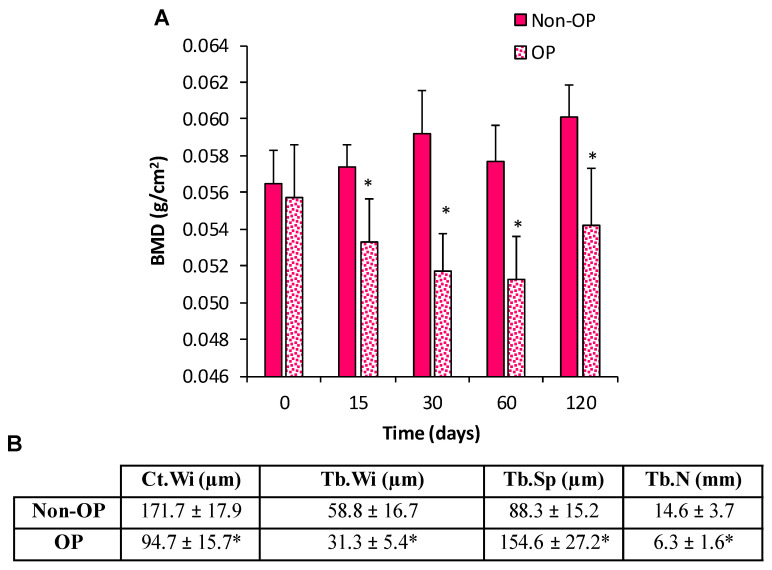
(**A**) Bone mineral density (BMD; g/cm^2^) values for control and OP animals at 0, 15, 30, 60, and 120 days of ovariectomy. (**B**) Histomophometric analysis in femurs measuring cortical bone width (Ct.Wi) and, in cancellous bone, trabeculae width (Tb.Wi), trabeculae separation (Tb.Sp), and trabeculae number (Tb.N). (*) denotes statistical significance differences compared with non-OP group (control) *p* < 0.001.

**Figure 5 pharmaceutics-13-00979-f005:**
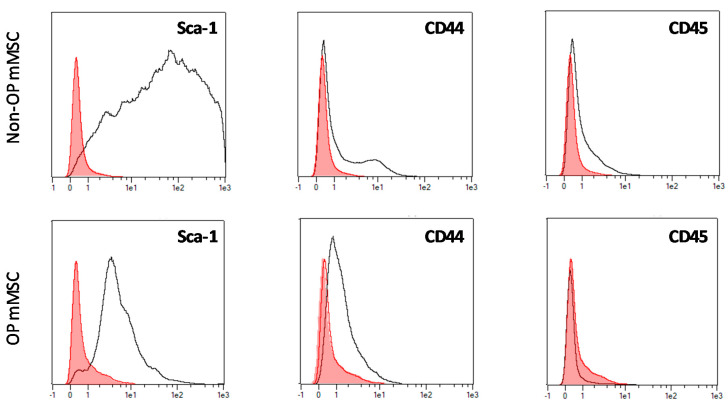
Cell surface markers of mouse mesenchymal stem cells extracted from healthy/control (non-OP mMSC) and osteoporotic mice (OP mMSC).

**Figure 6 pharmaceutics-13-00979-f006:**
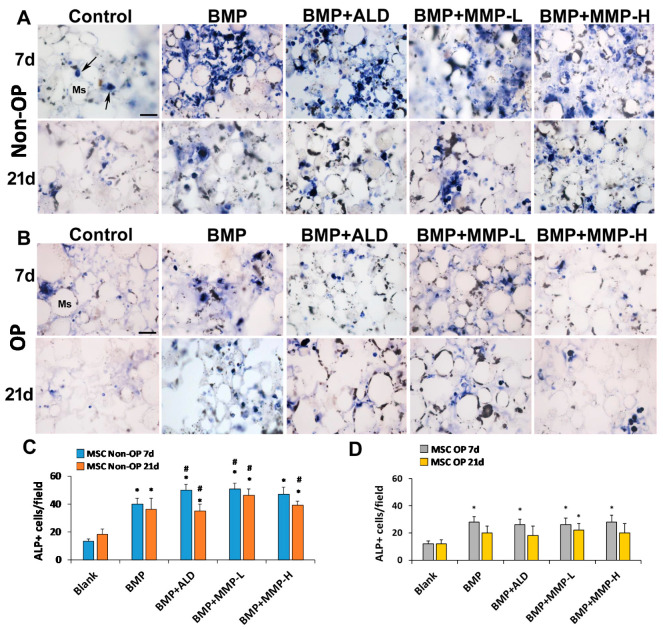
Panels (**A**,**B**) show representative images of mesenchymal stem cells from non-OP (**A**) and OP (**B**) animals cultured for 7 and 21 days and revealed for alkaline phosphatase activity. Histograms in (**C**,**D**) show the mean number of ALP positive cells per field at 400× magnification in non-OP and OP animals, respectively. The arrows in the upper left image indicate ALP-positive cells. Histograms represent mean ±SD values. (*) denotes statistical significant differences compared with control treatment at the corresponding time point. (#) denotes statistical significant differences compared with BMP treatment at the corresponding time point. Ms: microspheres. Scale bar = 30 µm.

**Figure 7 pharmaceutics-13-00979-f007:**
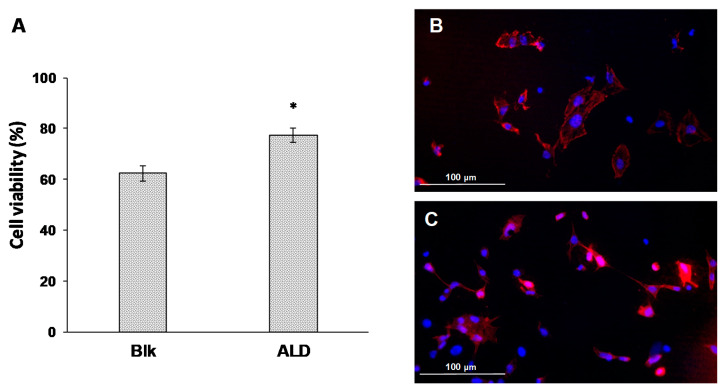
(**A**) Cell viability relative to control (polystyrene plate) of OP mMSC cultured on plasma-treated electrospun meshes containing covalently linked ALD or not and cell morphology of OP mMSC cultured on (**B**) ALD-linked electrospun meshes or (**C**) blk electrospun meshes. Blank (blk) stands for meshes without ALD. (*) denotes statistical significance differences compared with non-OP group (control) *p* < 0.001.

**Figure 8 pharmaceutics-13-00979-f008:**
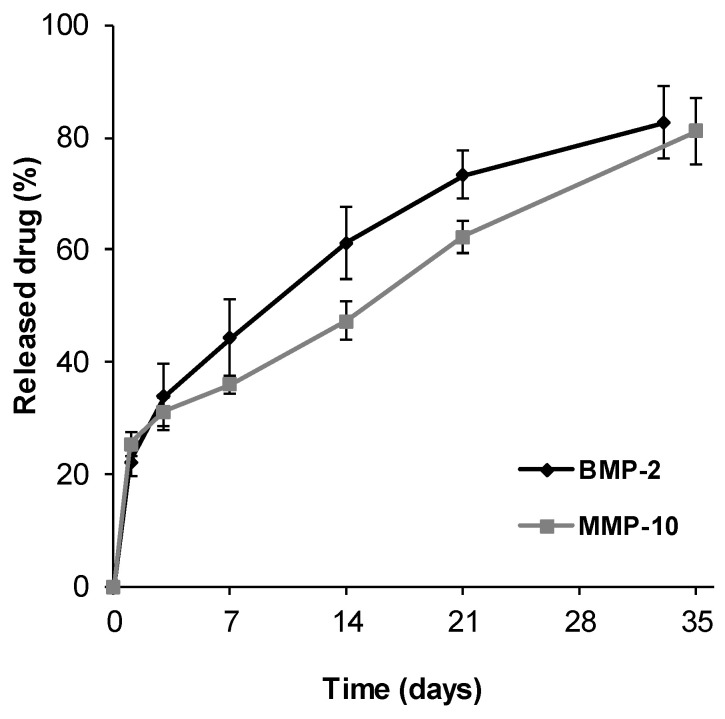
In vivo BMP-2 and MMP-10 release at different time points after implantation of the developed scaffolds on control animals. Release profiles were obtained using radiolabeled BMP-2 and MMP-10.

**Figure 9 pharmaceutics-13-00979-f009:**
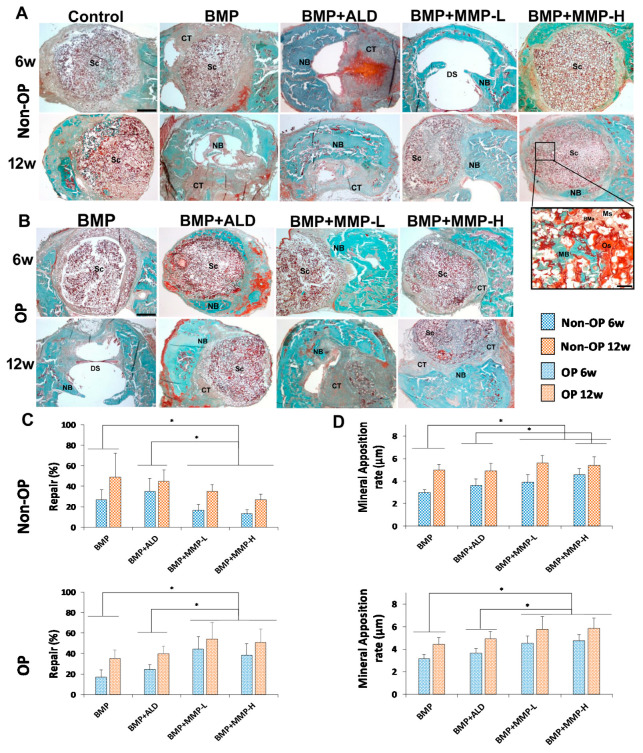
Representative panoramic images of the bone defect in the different experimental groups of non-OP (**A**) and OP (**B**) animals, showing the reparative response at 6 and 12 weeks post-implantation. Insert: Detail corresponding to the BMP + MMP-H group that shows active bone neoformation in the peripheral area of the scaffold (**C**,**D**). Histomorphometric analysis showing the percent of repair (**C**) and the mineral apposition rate (MAR) (**D**), six (blue) and twelve (orange) weeks post-implantation, in the different experimental groups of non-OP (squares) and OP (points) animals. Histograms represent mean ±SD values. The identical letter/symbol on different bars indicates significant differences. BMa: Bone marrow, CT: connective tissue, DS: defect site, MB: mineral bone, Ms: microspheres, NB: new bone, Os: osteoid, Sc: scaffold. Scale bars: A and B: 1 mm, inset: 80 µm. (*) denotes statistical significance differences between groups.

**Table 1 pharmaceutics-13-00979-t001:** Developed PLGA microspheres and drug amounts used per batch.

Microspheres Type	BMP-2 (µg)	ALD (µg)	MMP-10 (µg)
Blank PLGA	-	-	-
PLGA-BMP-2	35 µg	-	-
PLGA/Chitosan-BMP-2 + ALD	35 µg	75 µg	-
PLGA-BMP-2 + Low MMP-10	35 µg	-	2.1 µg
PLGA-BMP-2 + High MMP-10	35 µg	-	8.5 µg

**Table 2 pharmaceutics-13-00979-t002:** Developed cross-linked chitosan foams and their correspondent final drug amount adjusted by microsphere drug loading.

Nomenclature	Loaded Molecules (Dose)	Used Microspheres
Control foam	-	Blank PLGA
BMP-2 foam	BMP-2 (600 ng)	PLGA-BMP-2
BMP-2 + ALD foam	BMP-2 (600 ng) + ALD (75 µg)	PLGA/Chitosan-BMP-2 + ALD
BMP-2 + Low MMP-10 foam	BMP-2 (600 ng) + MMP-10 (30 ng)	PLGA-BMP-2 + Low MMP-10
BMP-2 + High MMP-10 foam	BMP-2 (600 ng) + MMP-10 (120 ng)	PLGA-BMP-2 + High MMP-10

**Table 3 pharmaceutics-13-00979-t003:** Treatment groups for the in vivo assessment of scaffold performance. The same treatment groups were used for control mice (non-OP group) and osteoporotic mice (OP group).

	Sandwich-Like Scaffolds
**Treatment Group**	**Chitosan Foam** 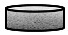	**Electrospun Mesh** 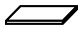
BMP	BMP-2 foam	Blank mesh
BMP + ALD	BMP-2 + ALD foam	ALD-mesh
BMP + MMP-L	BMP-2 + Low MMP-10 foam	Blank mesh
BMP + MMP-H	BMP-2 + High MMP-10 foam	Blank mesh
	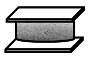

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
