# Peer review of "The Bone Regeneration Capacity of BMP-2 + MMP-10 Loaded Scaffolds Depends on the Tissue Status"

_pharmaceutics, 2021, doi:10.3390/pharmaceutics13070979_

Round 1

Reviewer 1 Report

Thanks for submitting this manuscript, which is evaluated the bone regeneration capacity of BMP2+MMP10 loaded scaffolds.

I have carefully read your manuscript with great interest.

I think that it should sound very interesting for readers and this paper overall well written.

This study is well designed and conducted.

I have a few minor comments.

In Figure 4, Authors need to clarify the abbreviation for Tb in figure legend. May be readers cannot distinguish the Tb.Wi, Tb. Sp, and Tb.N.

Ex) Trabecular width (Tb. Wi)

In section 3.3, authors describe the results of cytometry too simply. It is necessary to further explain the results. Because Sca-1 and CD44 expression of Non-OP mMSC show 10 times more distribution than the OP mMSC. If you can describe, it need to add the discussion part about this point. Is it possible that the positive effect of the BMP-2 + MMP-10 combination on the OP model can be attributed to the difference between OP-like mMSCs and healthy mMSCs?

Author Response

We would like to thank the reviewer for the careful reading of the manuscript. We have made several modifications to the manuscript and figures to address the reviewer’s concerns. Detailed responses to individual reviewer comments are given below in red. Corresponding alterations to the text have been highlighted in yellow in the manuscript.

Reviewer 

Thanks for submitting this manuscript, which is evaluated the bone regeneration capacity of BMP2+MMP10 loaded scaffolds.

I have carefully read your manuscript with great interest.

I think that it should sound very interesting for readers and this paper overall well written.

This study is well designed and conducted.

I have a few minor comments.

In Figure 4, Authors need to clarify the abbreviation for Tb in figure legend. May be readers cannot distinguish the Tb.Wi, Tb. Sp, and Tb.N. Ex) Trabecular width (Tb. Wi).

Figure 4 legend has been changed to the following: “A) Bone mineral density (BMD; g/cm2) values for control and OP animal at 0, 15, 30, 60 and 120 days of ovariectomy. B) Histomophometric analysis in femurs measuring cortical bone width (Ct.Wi) and, in cancellous bone, trabeculae width (Tb.Wi), trabeculae separation (Tb.Sp) and trabeculae number (Tb.N). (*) denotes statistical significance differences compared to Non-OP group (control) p < 0.001.”

In section 3.3, authors describe the results of cytometry too simply. It is necessary to further explain the results. Because Sca-1 and CD44 expression of Non-OP mMSC show 10 times more distribution than the OP mMSC. If you can describe, it need to add the discussion part about this point. Is it possible that the positive effect of the BMP-2 + MMP-10 combination on the OP model can be attributed to the difference between OP-like mMSCs and healthy mMSCs?

We thank the reviewer for the comment, indeed, the distributions of Sca-1+ cells and CD44+ cells are slightly different between Non-OP mMSC and OP mMSC. These differences could be associated to the disease instauration and, therefore, in some extend could affect the response to BMP-2+MMP-10. To clarify this point, the following paragraph has been added to the discussion section: “Cell populations were characterized by flow cytometry showing different profiles for Sca-1 and CD44 expression. Sca-1 deficient mice have been previously reported to undergo osteoporosis and are characterized by a decreased self-renewal of osteogenic stem cells [48-50]. In fact, the administration of Sca-1 + sorted MSC to osteoporotic animals was able to improve bone mineral density [51]. On the other hand, similarly to Sca-1, CD44 seemed to be decreased in OP mMSC. The absence of this cell surface receptor has been correlated to increased osteoclastogenesis and bone loss under inflammatory environments in mice [52]. Moreover, CD44 has been described as crucial for MSC migration, a cell function reported to be decreased in OP mMSC [53, 54]. Altogether, the modified positive expression of Sca-1 and CD44 observed for OP mMSC (Figure 5) could be associated to the disease instauration and the associated lower MSC migration, proliferation and differentiation capacity [54, 47]”.

Reviewer 2 Report

This is a manuscript on the synergistic effects of BMP-2 and MMP-10 on porotic bone. Authors employed the “sandwich-like” structure. Therapeutic molecules were encapsulated into PLGA spheres, and then embedded in chitosan foams, and two meshes were placed on top and bottom of the foam. A lot of materials, all of them seemed based on previous works, were involved in this study which unfortunately makes the whole story a little bit vague. However, authors did a lot of experiments to examine in vitro and in vivo behavior and performance of the composites, which was positive aspect of this manuscript. There was a discrepancy in the responsiveness to BMP-2&MMP-10 between non-porotic and porotic bones. This is an interesting study and may be worth publication but they need to fix minor issues shown below.

Line 31

Please spell out PLGA when it appears first.

Line 282

Authors illustrate the timeline of the OVX, steroid administration, BMD measurements, and surgery. How old were they at OVX surgery?

Line 287

Which part of the mouse was used for BMD analysis?

Line 373

Again, authors should describe the timeline. It looks like the OVX, steroid administration, and BMD measurements up to 120 days cannot be finished by 4 months.

Line 400-417

How many mice were used for this experiment?

Authors also need to describe end point of this experiment. When and how were they euthanized?

Figure1.

Please add scale bars.

Figure 7

Please describe what Blk stands for. Please mention figure 7B and 7C in figure legends.

Scale bars are needed.

Author Response

We would like to thank the reviewer for the careful reading of the manuscript. We have made several modifications to the manuscript and added a new supplementary figure to address the reviewer’s concerns. Detailed responses to individual reviewer comments are given below in red. Corresponding alterations to the text have been highlighted in yellow in the manuscript.

Reviewer 

This is a manuscript on the synergistic effects of BMP-2 and MMP-10 on porotic bone. Authors employed the “sandwich-like” structure. Therapeutic molecules were encapsulated into PLGA spheres, and then embedded in chitosan foams, and two meshes were placed on top and bottom of the foam. A lot of materials, all of them seemed based on previous works, were involved in this study which unfortunately makes the whole story a little bit vague. However, authors did a lot of experiments to examine in vitro and in vivo behavior and performance of the composites, which was positive aspect of this manuscript. There was a discrepancy in the responsiveness to BMP-2&MMP-10 between non-porotic and porotic bones. This is an interesting study and may be worth publication but they need to fix minor issues shown below.

Line 31

Please spell out PLGA when it appears first.

PLGA has been spelled out in the abstract and in the materials and methods section. Please check the highlighted text in yellow in the main manuscript.

Line 282

Authors illustrate the timeline of the OVX, steroid administration, BMD measurements, and surgery. How old were they at OVX surgery?

We thank the reviewer for the comment. The description of the OVX procedure has been changed to the following: “The osteoporosis mice model was obtained by a surgical procedure consisting of a dorsal incision ovariectomy (OVX) under inhaled anesthesia (isofluorane, ISOFLO®, Ab-bott Laboratories) on 16-weeks old FVB mice”.

Line 287

Which part of the mouse was used for BMD analysis?

Bone mineral density values were obtained from the whole animal. This point has been clarified in the main manuscript. Please check the highlighted text in yellow.

Line 373

Again, authors should describe the timeline. It looks like the OVX, steroid administration, and BMD measurements up to 120 days cannot be finished by 4 months.

 An additional Figure has been added on the supplementary materials section to clarify this point. As shown on the timeline summarized in the new Supplementary Figure 1, BMD measurements stopped before the implantation of the scaffolds as the data already indicated a significant decrease in BMD for OP mice when compared to Non-OP.

Line 400-417

How many mice were used for this experiment?

In total, 90 mice were used for the “in vivo” assessment of scaffold performance. From those 50 were Non-OP animals and 40 OP. This information has been clarified on the manuscript.

Authors also need to describe end point of this experiment. When and how were they euthanized?

Two endpoints were tested 6 and 12 weeks post-implantation, 5 animals of each group have been used for each time point (45 animals each). Animals were euthanized by CO2 inhalation. This information has been added to the materials and methods section. Please check the highlighted text in yellow.

Figure1.

Please add scale bars.

Figure 1 describes the NMR data, we think the reviewer is asking for scale bars in Figure 2 instead of 1. A new Figure 2 with the adequate scale bars has been added.

Figure 7

Please describe what Blk stands for. Please mention figure 7B and 7C in figure legends.

Scale bars are needed.

Both Figure 7 and Figure legend 7 have been changed to address the reviewer comments to the following: "Figure 7. A) Cell viability relative to control (Polystyrene plate) of OP mMSC cultured on plasma-treated electrospun meshes containing or not covalently linked ALD and Cell morphology of OP mMSC cultured on B) ALD-linked electrospun meshes or C) Blk electrospun meshes. Blank (Blk) stands for meshes without ALD. (*) denotes statistical significance differences compared to Non-OP group (control) p< 0.001".

Round 2

Reviewer 2 Report

All the issues I raised have been addressed appropriately.